# Based on the Spatial Multi-Scale Habitat Model, the Response of Habitat Suitability of Purpleback Flying Squid (*Sthenoteuthis oualaniensis*) to Sea Surface Temperature Variations in the Nansha Offshore Area, South China Sea

**DOI:** 10.3390/biology14060684

**Published:** 2025-06-12

**Authors:** Xue Feng, Xiaofan Hong, Zuozhi Chen, Jiangtao Fan

**Affiliations:** 1South China Sea Fisheries Research Institute, Guangzhou 510300, China; fengxue@scsfri.ac.cn (X.F.); zzchen2000@163.com (Z.C.); 2Key Laboratory for Sustainable Utilization of Open-Sea Fishery, Ministry of Agriculture and Rural Affairs, Guangzhou 510300, China; 3Zhangzhou Talent Development Group, Zhangzhou 363000, China; jyhdhkhxf114209@gmail.com; 4Southern Marine Science and Engineering Guangdong Laboratory (Guangzhou), Guangzhou 511458, China

**Keywords:** spatial multi-scale habitat model, habitat suitability, *Sthenoteuthis oualaniensis*, light falling-net fishing vessels, sea surface temperature, South China Sea

## Abstract

The purpleback flying squid (*Sthenoteuthis oualaniensis*) is a key cephalopod species in the South China Sea, gaining increasing commercial importance. In response to climate-induced shifts in marine ecosystems, this study employed habitat suitability index (HSI) models across six spatiotemporal scales to identify the optimal modeling resolution and to assess the squid’s habitat response to variations in sea surface temperature (SST). Analysis of fishery and environmental data from 2013 to 2017 revealed the highest model accuracy (84.02%) at a spatial resolution of 0.5° × 0.5° and a seasonal temporal resolution. The squid’s habitat proved highly sensitive to SST anomalies exceeding ±1 °C, resulting in the complete loss of suitable habitats. These findings offer scientific insights for sustainable resource management and climate-adaptive strategies in squid fisheries.

## 1. Introduction

The purpleback flying squid (*Sthenoteuthis oualaniensis*) is widely distributed across equatorial and subtropical waters of the Indian and Pacific Oceans, particularly in the South China Sea and the northwestern Indian Ocean [1]. In recent years, the squid fishery industry in the South China Sea has experienced rapid development, with annual catches ranging from 1.3 to 2 million tons [2]. Like other cephalopods, *S. oualaniensis* represents a highly promising source of marine protein and plays a critical role in the marine food web. It directly or indirectly influences the population dynamics of other marine organisms and occupies an essential ecological niche.

The purpleback flying squid is a crucial component of offshore fisheries in the South China Sea. Changes in its population directly affect regional fishery output and the livelihood of fishers. Global warming is expected to significantly alter the marine environment, which in turn may affect the distribution and abundance of marine cephalopods. Numerous studies have shown that the abundance and spatial distribution of cephalopods are closely linked to marine environmental conditions at various spatial and temporal scales.

The Habitat Suitability Index (HSI) model is a widely used method to evaluate the environmental preferences and habitat suitability of marine organisms. It is based on the assumption that habitat suitability can be determined if the biological responses of a species to environmental variables in a given area are known [3]. However, the relationship between fishery resources and the marine environment is complex and dynamic. Importantly, the spatial and temporal scales adopted during modeling significantly affect the outcomes [4]. Thus, selecting an appropriate spatial–temporal scale is crucial for drawing reliable conclusions.

To date, many researchers have studied the biology, population structure, spatial–temporal distribution, and environmental responses of the purpleback flying squid [5,6,7,8,9]. However, most studies have been conducted at specific spatial and temporal scales, with limited comparative analysis across multiple scales. Moreover, by correlating species occurrence with environmental variables, the HSI model can not only produce present-day habitat suitability maps but also estimate potential future distributions under various climate scenarios [10].

Although some preliminary studies have explored how environmental changes affect squid resources and fishing grounds, it remains unclear how climate-induced ocean warming influences the spatial and temporal distribution of *S. oualaniensis* habitats in the South China Sea and whether the extent of this influence is linked to habitat change.

In this study, we utilized Beidou satellite-based vessel monitoring system (VMS) data from light falling-net fishing vessels operating in the Nansha offshore area from 2013 to 2017 to develop HSI models at six different spatial–temporal resolutions. Acoustic survey data from 2013, 2016, and 2017 were used for model validation. Additionally, three SST variation scenarios (±0.2 °C, ±0.5 °C, and ±1 °C) were applied to assess the sensitivity of squid habitat to ocean temperature changes. The objectives of this research were as follows: (1) quantify the relationship between *S. oualaniensis* habitat suitability and SST, and (2) evaluate the effects of SST changes on squid habitat distribution in the Nansha Sea. The results aim to provide a scientific basis for the rational management and sustainable development of squid resources.

## 2. Materials and Methods

### 2.1. Data Sources

The purpleback flying squid is the primary target species of light falling-net fishing vessels operating in the Nansha offshore area of the South China Sea. Therefore, the number and distribution of these vessels are indicative of squid habitat preference and resource utilization. The spatial and temporal patterns of vessel activity reflect the distribution of squid resources and fishing grounds.

In this study, the number of light falling-net fishing vessels was used as a proxy for the quality of the fishing ground. Fishery data were obtained from the South China Sea Strategic Research Center for the period 2013–2017. These data included vessel location, operation time, and total vessel count. The temporal and spatial resolutions were 0.1° × 0.1°, covering the primary fishing zones between 5–16 °N and 109–119 °E (Figure 1).

Marine environmental data were sourced from the National Oceanic and Atmospheric Administration (NOAA) CoastWatch (https://coastwatch.pfeg.noaa.gov (accessed on 1 May 2025)). Three environmental variables were selected: sea surface temperature (SST), sea surface temperature anomaly (SSTA), and chlorophyll-a concentration (CHL), all with a spatial resolution of 0.1° × 0.1°. Daily environmental datasets were used, and if values were missing for specific grid points, Kriging interpolation was applied to estimate the missing data.

### 2.2. CPUE Calculation

Based on the Beidou satellite position data from light falling-net fishing vessels in the Nansha region, the catch per unit effort (CPUE) was calculated to represent the relative abundance of *S. oualaniensis*. The CPUE formula was as follows:(1)$${\mathrm{CPUE}}_{i}=\frac{{\mathrm{BOAT}}_{i}}{{\mathrm{BOAT}}_{\max}}$$
where ${\mathrm{BOAT}}_{i}$ is the number of vessels in the grid cell i for a given season or month, and ${\mathrm{BOAT}}_{\max}$ is the maximum vessel number within the same temporal scale.

### 2.3. Construction of the HSI Model

#### 2.3.1. Data Processing

To investigate the effects of different spatial and temporal scales on squid habitat modeling, six combinations were evaluated. The temporal resolutions were monthly and seasonal, and the spatial resolutions were 0.1°, 0.5°, and 1°, resulting in the combinations shown in Table 1. ArcGIS 10.8 was used to grid and align both fishery and environmental datasets according to each spatiotemporal scheme.

#### 2.3.2. Model Construction

Suitability index (SI) models based on CPUE were constructed for each spatiotemporal scheme. The SI for each environmental factor was calculated using the following:(2)$${\mathrm{SI}}_{i}=\frac{{\mathrm{CPUE}}_{i}}{{\mathrm{CPUE}}_{i,\max}}$$
where ${\mathrm{SI}}_{i}$ is the suitability index for a specific environmental range i, ${\mathrm{CPUE}}_{i}$ is the total CPUE for that range, and ${\mathrm{CPUE}}_{i,max}$ is the maximum CPUE across all ranges under the given spatiotemporal scenario. SI values range from 0 (unsuitable) to 1 (highly suitable) [11]. Univariate nonlinear regression was applied to model the relationship between SI and the environmental variables (SST, SSTA, and CHL), using DPS (v9.50) software.

#### 2.3.3. Model Validation and Comparison

The habitat suitability index (HSI) was computed as the geometric mean of SI values for SST, SSTA, and CHL:(3)$$\mathrm{HSI}=\sqrt[{3}]{{\mathrm{SI}}_{\mathrm{SST}}\;\times \;{\mathrm{SI}}_{\mathrm{SSTA}}\;\times \;{\mathrm{SI}}_{\mathrm{CHL}}}$$

Model accuracy was evaluated by comparing predicted HSI values with actual fishery data. Predictions with an absolute difference of less than 0.3 from observed values were considered accurate [12]. VMS data from 2014 to 2017 were used for model construction and comparison, while catch data from 2013, 2016, and 2017 were used for validation.

### 2.4. Suitability Index (SI) Modeling for SST

#### 2.4.1. Data Processing

The CPUE-based SI model for SST was constructed using VMS data from four representative months (March (spring), June (summer), September (autumn), and December (winter)) from 2013 to 2017. The spatial resolution was 0.1° × 0.1°, and the study area covered 6.4–12° N and 109–117° E.

#### 2.4.2. Model Construction

An empirical HSI model was used to analyze the influence of sea surface temperature change on the habitat suitability of purpleback flying squid in the Nansha offshore area. Habitat suitability is represented by a predicted HSI value of 0–1, that is, the probability that the habitat is suitable for the study objective [13]. A value of “0” indicates a completely unsuitable habitat, and a value of “1” indicates an optimal habitat. The HSI value was divided into four grades according to the equal division method, namely, 0–0.25 (unsuitable), 0.25–0.5 (generally suitable), 0.5–0.75 (suitable), and 0.5–1 (most suitable) [12]. Purpleback flying squid is the main catch of light falling-net fishing in Nansha [14], and the dense area of vessels means that it is suitable for the growth and survival of purpleback flying squid.

A single-factor SI model was made with SST, using CPUE in four quarters from 2013 to 2017. The range of SI was from 0 to 1. When the CPUE in the region was maximum, SI equaled 1. When CPUE was 0, SI equaled 0 [15,16]. The SI model was established according to CPUE, as follows:(4)$${\mathrm{SI}}_{i}=\frac{{\mathrm{CPUE}}_{i}}{{\mathrm{CPUE}}_{\max}}$$
where ${\mathrm{SI}}_{i}$ refers to the SI values computed based on CPUE in the interval i, ${\mathrm{CPUE}}_{i}$ refers to the total CPUE in the interval *i*, and ${\mathrm{CPUE}}_{\max}$ refers to the maximum CPUE in the i range.

The relationship between SI and SST was modeled using univariate nonlinear regression, with the following fitting equation:(5)$${\mathrm{SI}}_{\mathrm{SST}}=\exp\;(a\;\times \;{(\mathrm{SST}\;-\;b)}^{2})$$
where a and b are the model parameters estimated by the least-squares method to minimize the residual difference between the observed and predicted values. The value of ${\mathrm{SI}}_{\mathrm{SST}}$ ranges from 0 to 1.

#### 2.4.3. SST Variation Scenarios and Habitat Suitability Response

To assess how varying SST levels affect squid habitat distribution, the monthly mean SST (baseline climate conditions from 2014–2017) was adjusted by ±0.2 °C, ±0.5 °C, and ±1.0 °C. The corresponding changes in suitable (HSI > 0.5) and optimal (HSI > 0.75) habitat areas were analyzed seasonally in the Nansha offshore region.

## 3. Results

### 3.1. Seasonal and Monthly Distribution of Fishing Vessels

The number of active light falling-net fishing vessels from 2014 to 2017 was analyzed by season. The results showed that vessel counts were lowest in autumn (275) and highest in summer (1427), followed by spring (952) and winter (588). Monthly statistics revealed that June had the highest number of operating vessels (1056), accounting for 74.00% of the summer total. May also had relatively high activity, whereas other months saw substantially lower vessel numbers (Figure 2).

### 3.2. Model Accuracy Comparison Across Spatial–Temporal Scales

Based on SST, SSTA, and CHL, HSI models were constructed under six spatial–temporal scale combinations (Table 2). The accuracy of each model was evaluated by calculating the proportion of grid cells with less than 0.3 deviation between the predicted and observed HSI values.

In this study, the average accuracy of the HSI model at the same spatial–temporal resolution was used to screen the optimal model, and the acoustic survey data of purpleback flying squid sailing in 2013, 2016, and 2017 were cross-verified.

On the time scale of seasons, the average accuracy of the model corresponding to 0.1° × 0.1° spatial resolution was 66.692%, that of 0.5° × 0.5° spatial resolution was 84.023%, and that of 1° × 1° spatial resolution was 67.742%. When the month is taken as the time scale, the average accuracy of the model corresponding to 0.1° × 0.1° spatial resolution is 73.432%, the average accuracy of the model corresponding to 0.5° × 0.5° spatial resolution is 60.206%, and the average accuracy of the model corresponding to 1° × 1° spatial resolution is 76.069% (Table 3).

In summary, the model with the highest overall performance was the one using a 0.5° × 0.5° spatial resolution and seasonal temporal scale, which was thus selected as the optimal resolution for modeling squid habitat suitability in the Nansha offshore area.

### 3.3. Relationships Between CPUE and Environmental Variables

SI models for each environmental factor were fitted using CPUE data at the optimal spatiotemporal scale (0.5° × 0.5°, seasonal). Univariate nonlinear regression showed that all fits were statistically significant (*p* < 0.05), indicating good model reliability (Table 4).

The optimum value of each factor in each season is obtained when SI is close to 1, and the optimum value range is greater than or equal to 0.6. In spring, the optimal value of SST was 30.68 °C and the optimal range was 30.40~30.90 °C, the optimal value of SSTA was 1.054 °C and the optimal range was 0.84~1.27 °C, and the optimal value of CHL was 0.07 mg·m^−3^ and the optimal range was 0.06~0.08 mg·m^−3^. The optimal value of SST in summer was 30.19 °C, the optimal range was 30.05~30.35 °C, the optimal value of SSTA was 1.05 °C, the optimal range was 0.94~1.17 °C, the optimal value of CHL was 0.23 mg·m^−3^, and the optimal range was 0.21~0.25 mg·m^−3^. In autumn, the optimal value of SST was 29.03 °C, the optimal range was 28.90~29.15 °C, the optimal value of SSTA was 1.13 °C, the optimal range was 0.93~1.32 °C, the optimal value of CHL was 0.14 mg·m^−3^, and the optimal range was 0.09~0.19 mg·m^−3^. In winter, the optimal value of SST was 27.34 °C, the optimal range was 27.07~27.63 °C, the optimal value of SSTA was 0.88 °C, the optimal range was 0.58~1.18 °C, the optimal value of CHL was 0.19 mg·m^−3^, and the optimal range was 0.18~0.19 mg·m^−3^ (Figure 3).

### 3.4. Model Verification

Using the optimal spatial–temporal scale (0.5° × 0.5°, seasonal), the HSI model was validated with acoustic survey data from 2013, 2016, and 2017. The verification results showed that the accuracy of the HSI model in spring was 51.59%, the accuracy of the HSI model in summer was 63.11%, and the accuracy of the HSI model in autumn was 72.72% (due to the loud noise of the acoustic detection of navigation in winter, which greatly interfered with the results, it has not been included in the total results for the time being) (Table 5). This indicates that the spatiotemporal scale habitat model can predict the central fishing ground of purpleback flying squid in the Nansha offshore area well.

### 3.5. Suitable SST Ranges for the Purpleback Flying Squid

Using vessel distribution data and the HSI model, the SST suitability ranges for squid habitat were identified (Table 6, Figure 4).

The difference between the SST fitting value and the actual value was small (*p* < 0.01). The optimum SST values in spring, summer, autumn, and winter were 28.11 °C, 30.24 °C, 28.98 °C, and 28.79 °C, respectively.

### 3.6. Impact of SST Variation on Habitat Suitability

When SST increases by 0.2 °C, 0.5 °C, and 1 °C respectively, the habitat distribution changes as follows (Table 7 and Table 8, and Figure 5): The suitable habitat of purpleback flying squid in winter and spring is mainly distributed in the eastern region, and the suitable habitat migrates to the west in summer, while no suitable habitat appears in the autumn in the study area.

With the increase in SST, the habitat of purpleback flying squid in the Nansha offshore area changed greatly. Under the situation of SST + 0.2 °C, suitable habitat tends to expand in spring, summer, and autumn compared to habitat distribution during the period 2013−2017. There was no most suitable habitat area in autumn, but the most suitable habitat increased in other seasons. Under the situation of SST + 0.5 °C, the suitable habitat and the most suitable habitat area of autumn purpleback flying squid disappeared, and the suitable habitat in other seasons shrank. The area of suitable habitat increased in winter and spring, while the area of optimum habitat decreased in summer. Under the situation of SST + 1 °C, the habitat of purpleback flying squid in the Nansha offshore area changed the most. During each season, areas suitable for the purpleback flying squid completely disappear.

From the perspective of spatial distribution, in spring, with the increase in temperature, the suitable habitat gradually shifted to the northwest and finally disappeared. In summer, the suitable habitat is mainly distributed in the sea area around 8 °N, moving to the northeast and forming suitable habitat around 111 °E and 115 °E, respectively. As temperatures continue to rise, the suitable habitat shrinks to the west and tends to disappear. The spatial distribution trend of suitable habitat in winter and spring is similar, both of which shift from east to west, shrink, and finally disappear. In autumn, only under the condition of SST + 0.2 °C, 2.9% of suitable habitat was found in the central part of the study area, while no suitable habitat area was found under the other conditions of SST rise and original conditions.

With the decrease of SST, the suitable habitat in spring moved to the southwest, gradually contracted and finally tended to disappear. In winter and summer, with the decrease in temperature, the area of suitable habitat decreases significantly, and the distribution of suitable habitat shrinks based on the original location. In autumn, when the SST decreased by 0.2 °C and 0.5 °C, the area of suitable habitat increased by 36.73% and 53.36%, respectively. When the temperature is reduced by 1 °C, the suitable habitat area in autumn disappears. In most scenarios where SST dropped, there was no optimal habitat distribution area for the purpleback flying squid (Table 7 and Table 8, and Figure 6).

## 4. Discussion

In actual marine fisheries, fishing activities are conducted across various spatial and temporal scales, and fish population distributions respond accordingly [17]. However, most previous studies on squid habitat or fishing ground distribution have been conducted under a single, fixed spatial–temporal framework, without accounting for how scale itself might influence the outcome. Therefore, selecting an appropriate spatiotemporal scale is essential to ensure scientifically robust model predictions.

The population structure of *S*. *oualaniensis* is complex, with different groups exhibiting distinct migration directions and ranges. Generally, the species migrates from deep to shallow waters for reproduction and from shallow to deep waters in winter [1]. The spring and summer seasons represent peak fishing periods in the Nansha Sea, with well-established migratory routes forming prime fishing grounds [18].

In this study, we evaluated six habitat modeling schemes combining three spatial (0.1°, 0.5°, and 1°) and two temporal (monthly and seasonal) resolutions. Fishery and environmental data were processed under each scheme to examine how scale affected model accuracy. The results clearly showed that spatial–temporal scale significantly influenced habitat suitability modeling. Specifically, the model with a 0.5° × 0.5° spatial resolution and seasonal time scale had the highest predictive accuracy (84.02%), outperforming all other configurations. This optimal model effectively predicted the spatial distribution of suitable squid habitats.

To validate the model, predicted HSI values were compared with acoustic survey data. Higher concentrations of suitable habitat corresponded well with observed squid abundance (Figure 7), indicating that the model is a reliable tool for forecasting habitat distribution. Regions without fishing activity in the data are likely unexplored areas that could represent potential fishing grounds, warranting future investigation.

Fishing vessels were mainly concentrated in the northern Nansha offshore waters. Seasonally, they were densely distributed in the northwest during summer, shifted southeast in autumn, and returned northward in winter. These patterns are closely aligned with the acoustic detection results [5]. Previous studies have also suggested that fishing effort distribution is a more accurate indicator of habitat suitability than CPUE data alone [3]. Therefore, using vessel numbers as a proxy in this study provided a sound basis for HSI modeling.

Vessel activity peaked in summer and dropped to its lowest levels in autumn and winter. This pattern was influenced by the South China Sea fishing moratorium, the distribution of squid resources, and seasonal sea conditions [14,19,20]. The southeast monsoon in spring and southwest monsoon in summer cause strong upwelling in the northern Nansha waters, increasing nutrient availability and thus prey abundance for squid [21,22]. Additionally, during the June fishing moratorium north of 12 °N, vessels congregate in the southern Nansha area. After the moratorium ends in late August, vessels return north. Poor sea conditions in autumn and winter further limit fishing activity [14,19,21].

HSI modeling results revealed that seasonal changes in SST did not drastically alter habitat suitability, as the Nansha Sea is located in tropical waters with relatively stable temperatures. While temperature is a critical factor influencing cephalopod distribution, growth, and reproduction [17,23], SST variation in this region was not the main driver of seasonal changes in suitable habitat. This conclusion is supported by earlier studies on the environmental preferences of *S. oualaniensis* in the South China Sea, such as prey abundance [7,24].

However, the squid’s habitat proved highly sensitive to temperature anomalies. The species exhibits a narrow temperature tolerance range; thus, even small deviations in SST can lead to significant habitat contraction. In both +1 °C and −1 °C SST scenarios, suitable habitat areas were completely lost, indicating that the Nansha offshore area would become entirely unsuitable for squid survival and reproduction under such thermal stress. Cooling scenarios had more severe effects than warming, though slight warming did promote growth and habitat expansion to some extent. These results are consistent with prior studies identifying SST as the key factor influencing squid abundance in the South China Sea [25,26].

Other studies have shown that La Niña events can reduce squid recruitment by altering spawning conditions, whereas El Niño events may enhance recruitment [27]. Climate-driven changes in ocean conditions clearly affect squid resource availability and fishing ground dynamics. This study’s quantitative assessment of habitat responses to SST variations offers valuable insights into how squid populations might react to future climate changes.

While this study emphasizes the relationship between squid habitat suitability and SST, squid distribution is shaped by multiple interacting environmental and biological factors. Extensive past research indicates that many squid species inhabiting open waters exhibit seasonal migrations and significant spatiotemporal variations in their abundance [3,28,29]. It is important to note that squid distribution is influenced by a complex interplay of biotic and abiotic factors. While SST plays a dominant role, factors such as chlorophyll concentration, sea surface height, lunar phases, prey availability, and ocean dynamics also contribute. Future modeling efforts should integrate these variables and consider multi-model ensemble approaches to enhance prediction accuracy [30], particularly for fisheries requiring fine-scale forecasts.

## 5. Conclusions

In this study, an integrated HSI model was developed using light falling-net fishery data and three key environmental variables (SST, SSTA, and CHL) to determine the most appropriate spatial–temporal resolution for modeling the habitat of *S*. *oualaniensis* in the Nansha offshore area of the South China Sea. The analysis demonstrated that the combination of a 0.5° × 0.5° spatial resolution and seasonal temporal resolution yielded the highest model accuracy (84.02%). This confirms that the selection of spatial–temporal scale significantly affects the reliability of habitat modeling and that scale optimization is essential for accurately identifying core fishing grounds. Scenario simulations of SST variation revealed that both warming and cooling by ±1 °C resulted in the complete loss of suitable habitats for the purpleback flying squid. Mild warming scenarios (e.g., +0.2 °C) slightly expanded suitable habitat areas in some seasons, whereas cooling generally had more adverse effects on habitat extent and quality. These results indicate a narrow optimal temperature range for squid habitat in the region, highlighting the species’ vulnerability to climate-induced thermal changes.

Moving forward, it is crucial to account for scale-dependent variability and select appropriate spatial–temporal resolutions when constructing habitat models. Moreover, since the marine environment is governed by the interaction between multiple dynamic factors, future models should incorporate additional variables and explore their coupled effects to improve the predictive capacity of habitat assessments. This approach will contribute to more accurate squid distribution forecasts and facilitate more adaptive, science-based fishery management strategies in the context of global climate change.

## Figures and Tables

**Figure 1 biology-14-00684-f001:**
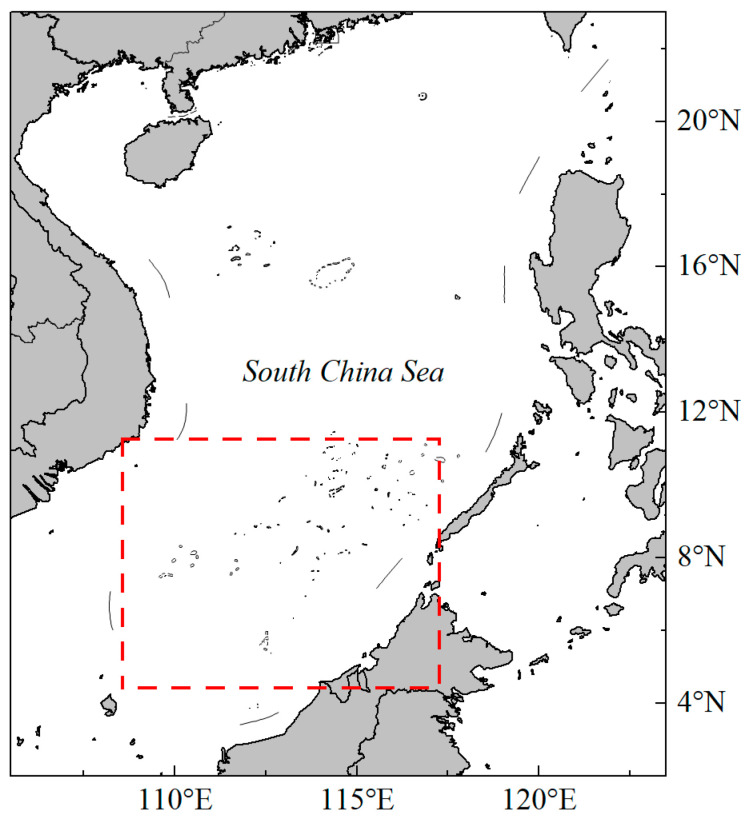
Research scope of the purpleback flying squid in the Nansha offshore area, South China Sea. The red box represents the fishing operation area for fishery data in 2013–2017.

**Figure 2 biology-14-00684-f002:**
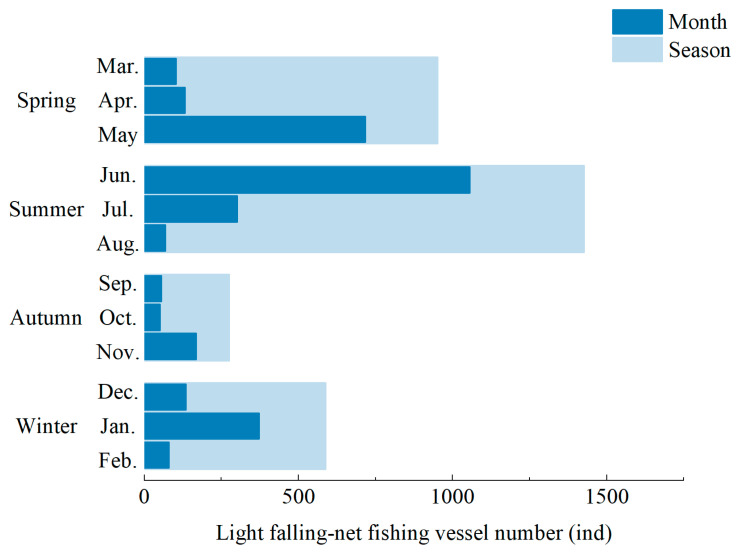
Distribution of light falling-net fishing vessel density in different seasons.

**Figure 3 biology-14-00684-f003:**
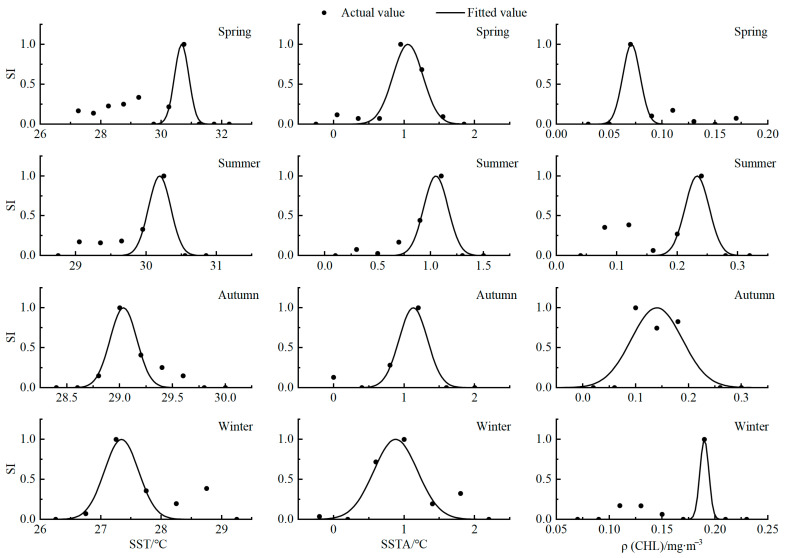
SI curves for each factor at optimal spatial and temporal scales.

**Figure 4 biology-14-00684-f004:**
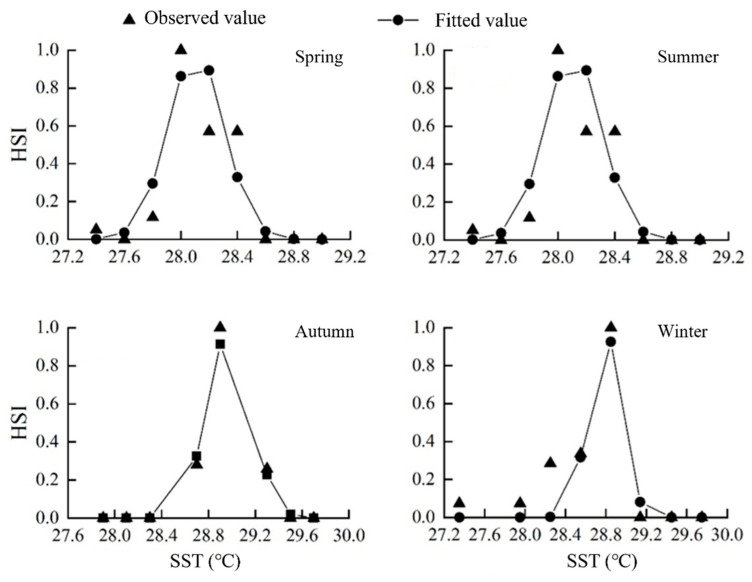
SST-based suitability curves.

**Figure 5 biology-14-00684-f005:**
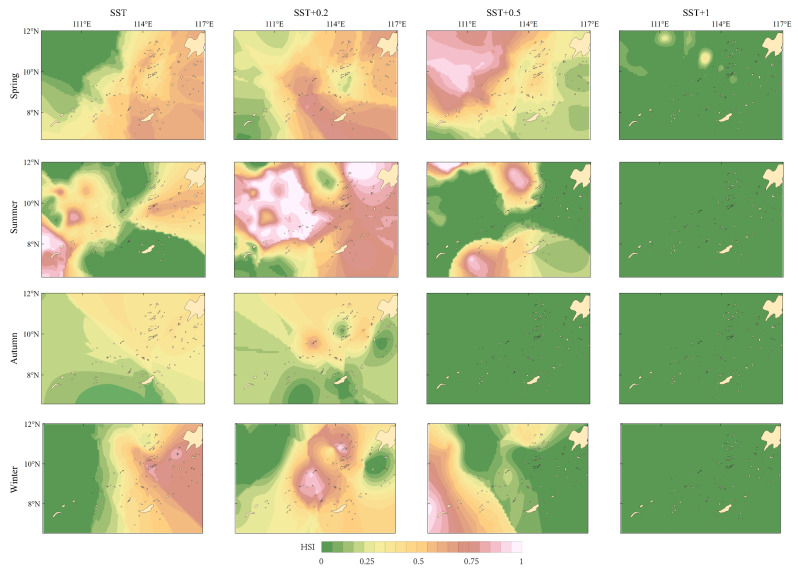
Variations in the habitat of purpleback flying squid in the Nansha offshore area when the sea surface temperature increases by 0.2 °C, 0.5 °C, and 1 °C.

**Figure 6 biology-14-00684-f006:**
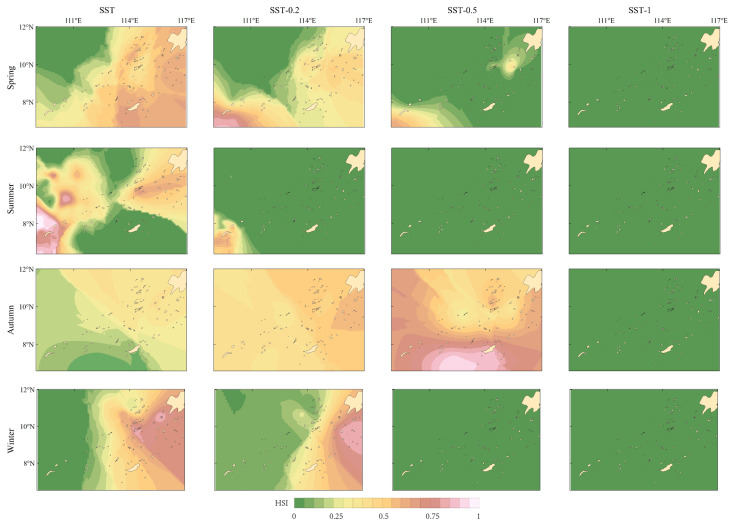
Variations in the habitat of purpleback flying squid in the Nansha offshore sea when the sea surface temperature decreases by 0.2 °C, 0.5 °C, and 1 °C.

**Figure 7 biology-14-00684-f007:**
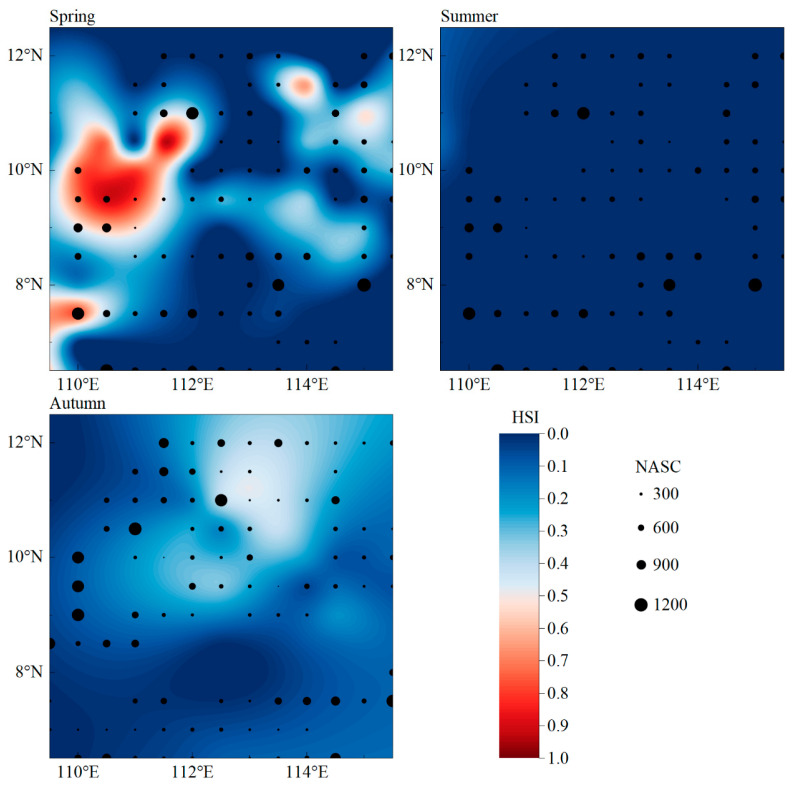
Schematic diagram of model validation.

**Table 1 biology-14-00684-t001:** Habitat modeling schemes at various spatial and temporal scales.

Scheme	Spatial Resolution	Temporal Resolution
1	0.1° × 0.1°	Season
2	0.1° × 0.1°	Month
3	0.5° × 0.5°	Season
4	0.5° × 0.5°	Month
5	1° × 1°	Season
6	1° × 1°	Month

**Table 2 biology-14-00684-t002:** Seasonal accuracy statistics of HSI models at three spatial scales.

Resolution		Spring	Summer	Autumn	Winter	Average Accuracy (%)
0.1° × 0.1°	Correct number	153	288	89	110	66.692
	Total	330	555	95	147	
	Accuracy (%)	46.364	51.892	93.684	74.830	
0.5° × 0.5°	Correct number	88	113	40	46	84.023
	Total	119	116	55	50	
	Accuracy (%)	73.950	97.414	72.727	92.000	
1° × 1°	Correct number	62	52	11	43	67.742
	Total	84	71	42	44	
	Accuracy (%)	73.810	73.239	26.190	97.727	

**Table 3 biology-14-00684-t003:** Monthly accuracy statistics of HSI models at three spatial scales.

Month	0.1° × 0.1°			0.5° × 0.5°			1° × 1°		
	Correct Number	Total	Accuracy (%)	Correct Number	Total	Accuracy (%)	Correct Number	Total	Accuracy (%)
1	25	53	47.170	8	10	80.000	7	8	87.500
2	27	33	81.818	13	19	68.421	12	15	80.000
3	58	62	93.548	18	35	51.429	17	24	70.833
4	59	67	88.060	25	28	89.286	19	27	70.370
5	75	201	37.313	11	56	19.643	30	33	90.909
6	339	364	93.132	55	57	96.491	23	33	69.697
7	78	168	46.429	11	45	24.444	18	27	66.667
8	19	23	82.609	4	14	28.571	8	11	72.727
9	20	22	90.909	12	15	80.000	10	11	90.909
10	10	17	58.824	14	15	93.333	10	11	90.909
11	55	56	98.214	12	25	48.000	16	19	84.211
12	24	38	63.158	9	21	42.857	8	21	38.095
Average accuracy (%)			73.432			60.206			76.069

**Table 4 biology-14-00684-t004:** SI model for each factor at appropriate spatial and temporal scales.

Season	Environmental Factor	SI Model	F	*p*	R^2^
Spring	SST	X_2_ = EXP (−8.926 × (X_1_ − 30.676)^2^)	17.673	0.0023	0.663
	SSTA	X_2_ = EXP (−11.125 × (X_1_ − 1.054)^2^)	136.179	0.0000	0.958
	CHL	X_2_ = EXP (−7310.241 × (X_1_ − 0.071)^2^)	122.700	0.0000	0.953
Summer	SST	X_2_ = EXP (−20.191 × (X_1_ − 30.191)^2^)	40.673	0.0007	0.871
	SSTA	X_2_ = EXP (−37.899 × (X_1_ − 1.050)^2^)	98.626	0.0001	0.943
	CHL	X_2_ = EXP (−1261.593 × (X_1_ − 0.233)^2^)	11.014	0.0160	0.647
Autumn	SST	X_2_ = EXP (−31.082 × (X_1_ − 29.035)^2^)	66.288	0.0001	0.905
	SSTA	X_2_ = EXP (−12.429 × (X_1_ − 1.130)^2^)	118.575	0.0004	0.967
	CHL	X_2_ = EXP (−214.695 × (X_1_ − 0.140)^2^)	22.887	0.0050	0.821
Winter	SST	X_2_ = EXP (−6.265 × (X_1_ − 27.341)^2^)	14.512	0.0125	0.744
	SSTA	X_2_ = EXP (−5.198 × (X_1_ − 0.880)^2^)	34.320	0.0021	0.873
	CHL	X_2_ = EXP (−28,883.236 × (X_1_ − 0.190)^2^)	88.568	0.0000	0.927

**Table 5 biology-14-00684-t005:** Habitat model accuracy statistics of purpleback flying squid based on walk-around acoustic survey data at optimal spatial and temporal scales.

	Spring	Summer	Autumn
Correct number	65	65	120
Total	126	103	165
Accuracy	0.515873	0.631068	0.727273

**Table 6 biology-14-00684-t006:** Habitat suitability index model by season.

Season	HSI Models	*p*	R^2^
Spring	HSI = EXP (−12.95 × (SST − 28.11)^2^)	0.0001	0.815
Summer	HSI = EXP (−33.80 × (X1 − 30.24)^2^)	0	0.898
Autumn	HSI = EXP (−14.43 × (X1 − 28.98)^2^)	0	0.988
Winter	HSI = EXP (−20.24 × (SST − 28.79)^2^)	0.0002	0.880

**Table 7 biology-14-00684-t007:** Variation in area of more suitable habitat for purpleback flying squid (0.5 < HSI < 0.75).

Scenarios	March (Spring)			June (Summer)			Autumn (September)			Winter (December)		
	Area (km^2^)	Area Ratio (%)	Area Change (%)	Area (km^2^)	Area Ratio (%)	Area Change (%)	Area (km^2^)	Area Ratio (%)	Area Change (%)	Area (km^2^)	Area Ratio (%)	Area Change (%)
SST	218,833	41.16		67,568	12.70		0	0		159,199	29.93	
SST + 0.2 °C	254,126	47.78	16.13	147,437	27.72	118.21	11,024	2.07	2.07	121,093	22.78	−23.94
SST + 0.5 °C	70,948	13.36	−67.58	51,860	9.75	−23.25	0	0	0	72,780	13.69	−54.28
SST + 1 °C	0	0	−100	0	0	−100	0	0	0	0	0	−100
SST − 0.2 °C	28,816	5.41	−86.83	12309	2.31	−81.78	195,224	36.73	36.73	72,746	13.66	−54.31
SST − 0.5 °C	11,899	2.23	−94.56	0	0	−100	283,815	53.36	53.36	0	0	−100
SST − 1 °C	0	0	−100	0	0	−100	0	0	0	0	0	−100

**Table 8 biology-14-00684-t008:** Variations in optimal habitat area for the purpleback flying squid (HSI > 0.75).

Scenario	March (Spring)			June (Summer)			September (Autunm)			December (Winter)		
	Area (km^2^)	Area Ratio (%)	Area Variation (%)	Area (km^2^)	Area Ratio (%)	Area Variation (%)	Area (km^2^)	Area Ratio (%)	Area Variation (%)	Area (km^2^)	Area Ratio (%)	Area Variation (%)
SST	0	0		34,006	6.39		0	0	0	24,059	4.52	
SST + 0.2 °C	20,029	3.77	3.77	264,973	49.83	679.19	0	0	0	32,118	6.04	33.50
SST + 0.5 °C	131,369	24.74	24.74	31,428	5.91	−7.58	0	0	0	34,973	6.58	45.36
SST + 1 °C	0	0	0	0	0	−100	0	0	0	0	0.00	−100
SST − 0.2 °C	15,940	2.99	2.99	0	0	−100	0	0	0	34,672	6.51	44.11
SST − 0.5 °C	0	0	0	0	0	−100	100,861	18.96	18.96	0	0	−100
SST − 1 °C	0	0	0	0	0	−100	0	0	0	0	0	−100

## Data Availability

The datasets generated and analyzed during the current study are available from the corresponding author upon reasonable request.

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
