# Peer review of "Based on the Spatial Multi-Scale Habitat Model, the Response of Habitat Suitability of Purpleback Flying Squid (Sthenoteuthis oualaniensis) to Sea Surface Temperature Variations in the Nansha Offshore Area, South China Sea"

_biology, 2025, doi:10.3390/biology14060684_

Round 1
Reviewer 1 Report
Comments and Suggestions for Authors
This paper analyzes how the spatiotemporal scale affects the forecast results of fishing grounds and the impact of surface temperature changes on squid fishing grounds, which has certain research significance.But in terms of English writing proficiency, it needs to be strengthened and revised. There are also some other issues that need to be clarified:
1、The CPUE calculation method is somewhat biased, only using the ratio of the number of ships in the unit fishing area grid to the study area to represent the relative abundance of fishery resources, which is different from the actual fishery location and can only reflect the intensity of fishing vessel activity
2 Figure 2, 1-12 Suggest changing it to Jan. - Dec., which is an English month representation
3 Table 2 and Table 3 show that the accuracy of predicting fishing grounds at a grid scale of 0.1 and 1 degrees is exactly opposite on a quarterly scale to a monthly scale, while the accuracy of the 0.5 degree grid is also significantly opposite to the results of the other two scales. How can this phenomenon be explained? Normally, it is correct that the accuracy of forecasts on large and larger time scales is higher than that on smaller spatiotemporal scales. The finer the scale of the forecast results, the more difficult it is.
4 Quarterly scale fishery forecasting has no effect on actual fishery production, as it requires 3-day and 0.25 or 0.5 degree scale refined forecasting for fishery production. Therefore, in the discussion, it is best to combine the actual production needs of the fishery to make a suitable spatiotemporal scale recommendation for fishery forecasting.
Comments on the Quality of English Language
This paper analyzes how the spatiotemporal scale affects the forecast results of fishing grounds and the impact of surface temperature changes on squid fishing grounds, which has certain research significance.But in terms of English writing proficiency, it needs to be strengthened and revised. There are also some other issues that need to be clarified:
1、The CPUE calculation method is somewhat biased, only using the ratio of the number of ships in the unit fishing area grid to the study area to represent the relative abundance of fishery resources, which is different from the actual fishery location and can only reflect the intensity of fishing vessel activity
2 Figure 2, 1-12 Suggest changing it to Jan. - Dec., which is an English month representation
3 Table 2 and Table 3 show that the accuracy of predicting fishing grounds at a grid scale of 0.1 and 1 degrees is exactly opposite on a quarterly scale to a monthly scale, while the accuracy of the 0.5 degree grid is also significantly opposite to the results of the other two scales. How can this phenomenon be explained? Normally, it is correct that the accuracy of forecasts on large and larger time scales is higher than that on smaller spatiotemporal scales. The finer the scale of the forecast results, the more difficult it is.
4 Quarterly scale fishery forecasting has no effect on actual fishery production, as it requires 3-day and 0.25 or 0.5 degree scale refined forecasting for fishery production. Therefore, in the discussion, it is best to combine the actual production needs of the fishery to make a suitable spatiotemporal scale recommendation for fishery forecasting.
Author Response
Dear reviewer :
Thank you for your letter and the reviewers’ comments on our manuscript entitled “Based on the spatial multi-scale habitat model, the response of habitat suitability of purpleback flying squid (Sthenoteuthis oualaniensis) to sea surface temperature varieties in the Nansha offshore area, South China Sea” (Manuscript ID: biology-3630102). Those comments are very helpful for revising and improving our paper, as well as the important guiding significance to another research. We have studied the comments carefully and made corrections which we hope meet with approval. The main corrections are in the manuscript and the responds to the reviewers’ comments are as follows.
Specific comments
- The CPUE calculation method is somewhat biased, only using the ratio of the number of ships in the unit fishing area grid to the study area to represent the relative abundance of fishery resources, which is different from the actual fishery location and can only reflect the intensity of fishing vessel activity
Response: Your opinion is extremely valuable to us, and we have given serious consideration to your perspective. However, we believe that commercial fishing vessels are profit-driven and will only operate in proximity to fishing grounds. Existing studies indicate that when there is a sufficient number of fishing boats, their location is a more indicative measure of the quality of the fishing ground than factors such as power and other indicators.
- Figure 2, 1-12 Suggest changing it to Jan. - Dec., which is an English month representation
Response: Thank you very much for your suggestion. We have revised the figure to make its content more concise and accurate.
- Table 2 and Table 3 show that the accuracy of predicting fishing grounds at a grid scale of 0.1 and 1 degrees is exactly opposite on a quarterly scale to a monthly scale, while the accuracy of the 0.5 degree grid is also significantly opposite to the results of the other two scales. How can this phenomenon be explained? Normally, it is correct that the accuracy of forecasts on large and larger time scales is higher than that on smaller spatiotemporal scales. The finer the scale of the forecast results, the more difficult it is.
Response: Your question is very insightful. The authors of this manuscript have had further discussions and believe that remote sensing data on large time scales may obscure many specific characteristics of fishing grounds due to algorithmic issues, thereby reflecting more general features of the fishing areas. Only appropriate temporal and spatial scales can accurately and adequately express the spatiotemporal characteristics of fishing grounds. For instance, mid-scale eddies in the ocean have durations and spatial extents that correspond optimally with the best spatiotemporal resolution, and characteristics like eddies are crucial factors influencing the formation of fishing grounds and the levels of fishery yields.
- Quarterly scale fishery forecasting has no effect on actual fishery production, as it requires 3-day and 0.25 or 0.5 degree scale refined forecasting for fishery production. Therefore, in the discussion, it is best to combine the actual production needs of the fishery to make a suitable spatiotemporal scale recommendation for fishery forecasting.
Response: Your suggestions are a tremendous assistance in improving our manuscript. The content of our research leans more towards the trends of squid habitats under future climate change scenarios. We believe that against the backdrop of macro climate change, interannual climate characteristic differences will become increasingly pronounced. Understanding the overall trend of the relationship between squid habitats and climate characteristics will help assess future production prospects and enable proactive adjustments in production layouts according to anticipated habitat scenarios. At the same time, we also consider your suggestions to be very meaningful, and we have emphasized the limitations of this study in the discussion section.
Reviewer 2 Report
Comments and Suggestions for Authors
Manuscript ID: biology-3630102
Title: Based on the spatial multi-scale habitat model, the response of habitat suitability of purpleback flying squid (Sthenoteuthis oualaniensis) to sea surface temperature varieties in the Nansha offshore area, South China Sea
The MS is about the multi-scale habitat model of purpleback flying squid. I understand a good question behind this research which will pave the managers way! This is an interesting research subject in habitat modeling. However, I could not find novelty and may authors look through recent paper Han et al. (2023) which is highly similar to current research! So, the big question is how difference and what is new finding?
Why you select just SST and Chla? There are other environmental variables that have been investigated in previous studies.
In recent study, 0.25° × 0.25° was defined the best prediction performance. Why don’t tried this resolution?
I would like to suggest the author reconsider recent references and use more of recent papers.
Discussion needs to be revised and justify by results. I could not find strong discussion and compare with recent research. There are many papers regarding squid in South China Sea.
In overall, I am recommending the article for publishing in J of biology after Major revision.
Best Regards,
-------------------------------------
Han, H., Jiang, B., Shi, Y., Jiang, P., Zhang, H., Shang, C., ... & Xiang, D. (2023). Response of the Northwest Indian Ocean purpleback flying squid (Sthenoteuthis oualaniensis) fishing grounds to marine environmental changes and its prediction model construction based on multi-models and multi-spatial and temporal scales. Ecological Indicators, 154, 110809.
Author Response
Dear reviewer #2:
Thank you for your letter and the reviewers’ comments on our manuscript entitled “Based on the spatial multi-scale habitat model, the response of habitat suitability of purpleback flying squid (Sthenoteuthis oualaniensis) to sea surface temperature varieties in the Nansha offshore area, South China Sea” (Manuscript ID: biology-3630102). Those comments are very helpful for revising and improving our paper, as well as the important guiding significance to another research. We have studied the comments carefully and made corrections which we hope meet with approval. The main corrections are in the manuscript and the responds to the reviewers’ comments are as follows.
Specific comments
- So, the big question is how difference and what is new finding? Why you select just SST and Chla?There are other environmental variables that have been investigated in previous studies.
Response: We greatly appreciate your thorough review of the manuscript. We have carefully discussed the viewpoints you raised; however, we believe that the marine environment of the South China Sea is complex, with various currents, typhoons, and other marine events. The uniqueness of the marine environment makes the formation of fishing grounds more difficult to explain compared to other sea areas. The references cited focus on the open waters of the Indian Ocean, where changes in the marine environment are relatively predictable. We have selected only SST and Chla because these two environmental factors are the easiest to obtain, and various remote sensing data and in-situ measurements can effectively verify each other's accuracy and stability.
- In recent study, 0.25° × 0.25° was defined the best prediction performance. Why don’t tried this resolution?
Response: The question you raised is very pertinent. We discussed and concluded that the environmental characteristics of different marine areas vary, and the key environmental factors that shape fishing grounds are also not the same. The indicators used to assess the quality of fishing grounds largely determine the spatial resolution, and for commercial fisheries, both excessively high and excessively low spatial resolutions are detrimental to guiding fishing production.
- I would like to suggest the author reconsider recent references and use more of recent papers. Discussion needs to be revised and justify by results. I could not find strong discussion and compare with recent research. There are many papers regarding squid in South China Sea.
Response: Thank you very much for your thorough review of the contents of this manuscript. We have re-examined and revised the discussion section of the manuscript based on your suggestions and the recommended literature, adding comparisons of conclusions from related studies.
Round 2
Reviewer 2 Report
Comments and Suggestions for Authors
Dear Editor,
The manuscript revised properly by authors. In overall, I am recommending the article for publishing in J of biology.
Best Regards,